# No Room for Mistakes: The Impact of the Social Unconscious on Organizational Learning in Kazakhstan

Indira Kjellstrand  and Russ Vince * 

School of Management, University of Bath, Bath BA6 8FD, UK; indiratk@gmail.com
* Correspondence: r.vince@bath.ac.uk; Tel.: +44-1225-384-419

**Abstract:** The aim of this paper is to add to existing work on the theme of power, emotion, and organizational learning. The study was undertaken in Kazakhstan, where tensions between old and new regimes provide an environment that is rich in emotion and power/politics; and offer an opportunity to study the interplay between emotion and power during individual and organizational attempts to learn. The *social unconscious* is used as a conceptual frame to identify underlying dynamics that impact on organizational learning. The empirical study illustrates a social fantasy concerning the fear of mistakes and its consequences. This fantasy is sustained through blaming and punishing the people who make mistakes, and through feelings of internalised embarrassment and guilt that are enacted through interpersonal relations of shaming and being ashamed. Our contribution to knowledge arises from employing a concept (social unconscious) that has not been used to study organizational learning within a social and organizational context for organizational learning (Kazakhstan) that has not yet been studied. The practical purpose of this paper is to improve our knowledge of the social and political context of organizational learning in post-Soviet Kazakhstan through understanding unconscious dynamics that both inform and undermine attempts to learn.

**Keywords:** organizational learning; emotions; mistakes; the social unconscious; power; Kazakhstan

## 1. Introduction

The social context for this study of organizational learning is Kazakhstan, which continues to undergo transformation from a Soviet system with a planned economy towards a capitalist free market economy. This context is worthy of attention because elements of the old (Soviet) regime linger in organizations that are part of Kazakhstan's emergent market economy. The tension between old and new regimes provides an environment that is rich both in emotion and power/politics, and it offers an opportunity to study how the interplay between emotion and power affect the learning processes within and between individuals as well as organizational learning. Post-Soviet organizations are places where the old ways of running businesses paradoxically coexist with a free market economy. Even though post-Soviet organizations are run according to these new rules, the structural and social aspects of organizations are still informed by the Soviet past (Schwartz and McCann 2007; Schwartz 2004). The dynamics that inform this paradox are not only rational. There are also unconscious processes that inform collective behaviour and organising in modern Kazakhstan. Existing studies on organizations in Kazakhstan have reported two issues, namely, a 'no trust' culture (Minbaeva et al. 2007) and the passivity of people who tend to avoid taking initiative (Muratbekova-Touron 2002). In addition, the absence of efficient methods for employee development, such as a constructive feedback loop, were mentioned as another issue preventing learning. Organizations in Kazakhstan use 'attestatsia' (attestation), which supports the ability of management to check whether an employee meets the requirements of the position he or she is occupying (Minbaeva et al. 2007). In the literature of emotions,

power, and organizational learning, the above issues are connected to anxiety, blame, and lack of communication (Vince and Gabriel 2011; Vince and Saleem 2004). For instance, Vince and Saleem (2004) studied the interrelation between fear of mistakes and anxiety. They conclude that these emotions cause negative learning patterns such as cautious behaviour, self-protectiveness and, when things go wrong, blaming others. We think that a 'no trust' culture, peoples' passivity, and the tendency of the Kazakhstani management to check people instead of developing a constructive feedback method, may be linked to similar emotions and, consequently, had developed power relations negatively affecting organizational learning. Moreover, as we argue, such emotions and power relations may stem from unconscious defences (Vince and Gabriel 2011). We illustrate this by focusing on an overarching social and political dynamic which we refer to as 'no room for mistakes'.

Our aim is to add to existing work on the theme of power, emotion and organizational learning (Vince and Gabriel 2011; Vince 2001). Our contribution to knowledge arises from using the concept of the 'social unconscious' (Weinberg 2007) to identify key dynamics within the emotional and political context of organizational learning. Such dynamics provide insights into the possibilities and limitations of organizational learning in Kazakhstan. The 'social unconscious' provides us with a conceptual framework through which to explore the interface between the individual and the social. By doing so, we also respond to calls to address the impact of societal-environmental elements on organizational learning (Schilling and Kluge 2009). The 'social unconscious' helps us to illuminate how individuals unconsciously perform taken-for-granted, socially embedded habits along with organizational agendas. Our empirical investigation illustrates how people in organizations are caught up in the tension between the Soviet legacy and new market economy rules. The practical purpose of this paper is to improve our knowledge of the social and political context of organizational learning in post-Soviet Kazakhstan through understanding unconscious dynamics that both inform and undermine attempts to learn. We begin to articulate the problems and possibilities for both individual and organizational learning within this distinct social order, as well as identifying implications for practice.

The paper is organized in the following way. In the next section, we provide the conceptual background to our empirical study by discussing the literature on process-oriented organizational learning, organizational power relations, and the social unconscious. We build a conceptual framework through which to analyse organizational learning in Kazakhstan. We present our research design and methods, followed by an analysis of vignettes drawn from the data that relate to one aspect of the social unconscious (fear of mistakes). We discuss and develop the main dimensions of our study and, in the conclusion, draw out implications for research and practice.

## 2. Organizational Learning, Power and the Social Unconscious

A process oriented approach to organizational learning acknowledges the dynamic and social nature of learning (Crossan et al. 1999, 2011; Vince and Gabriel 2011). Learning occurs, or is resisted, through interactions and relations between individuals, and within and between groups (Brandi and Elkjaer 2011; Easterby-Smith and Lyles 2011; Crossan et al. 1999). This focus on organizational learning is not concerned with the identification of specific capabilities or performance but, rather, is focused on people's participation in everyday organizational life (Brandi and Elkjaer 2011). Organizational learning is viewed as more than an output. It extends into social processes of co-creating, knowing and learning. This approach seeks to reveal the intricacies and nonlinearity of any learning process within its distinctive context (Berends and Lammers 2010) and helps us to appreciate the complexity of the social processes in which individuals are embedded (Brandi and Elkjaer 2011).

A process oriented approach asks scholars to think of the individual as a person who is embedded in an organizational context. Learning in organizations is therefore understood as 'part of everyday organizational life and work. Learning cannot be avoided; it is not a choice for or against learning; learning is not restricted to taking place inside individuals' minds but as a process of participation and interaction' (Brandi and Elkjaer 2011, p. 28). Organizational learning is more than a rational, positive, conscious, and well thought-out process. For example, within organizations, individuals deal with their

everyday commitments and can learn from unintended actions (Vince and Gabriel 2011). However, various social forces are likely to impede learning (Berends and Lammers 2010) as people within organizations can become caught in internal, political dynamics that discourage learning (Vince and Saleem 2004).

From this perspective, organizational learning is interwoven with organizational power relations that can support and undermine people's ability to learn (Vince 2001; Vince and Gabriel 2011). Systemic power relations (Fleming and Spicer 2014) are built and supported by social interaction among people, they are the product of shared knowledge (Willmott 2013). This view of power in organizations probes the strategic use of people's autonomy, and moves from a consideration of official management commands and the idea of power as possession to a focus on power that is intimately bound up with people's effort, intentionality and agency in organizations (Creed et al. 2014; Fleming and Spicer 2014). In other words, it constitutes peoples' identities in organizations, 'so that actors manage themselves on the behalf of vested interests, often in the name of free self-expression, autonomy and career development' (Fleming and Spicer 2014, p. 36). From this perspective, power relations can be interpreted as 'value-imbued fantasies which act to represent particularities as universalities' (Willmott 2013, p. 55). By so doing, certain ways of thinking and acting become unquestioned aspects of everyday interactions and social expectations.

In this study, we further argue that broader societal relations that reflect our lived experience shape power relations in organizations. We specifically focus on the concept of the *social unconscious* which helps us to capture unconscious dynamics that inform and perpetuate social order. The social unconscious refers to a 'co-constructed shared unconscious of members of a certain social system such as community, society, nation or culture' that 'includes shared anxieties, fantasies, defences, myths, and memories' (Weinberg 2007, p. 312). Here the concept refers to shared and established fantasies[1] that are sustained and perpetuated by people in organizations. Like the individual unconscious, the social unconscious is 'out of space and timeless', and 'the members of a group are able to re-live and re-enact in the here-and-now relationships and pertinent emotions from the remote past' (Weinberg 2007, pp. 308–9). The difference of the social unconscious from other preconscious, subconscious, and social elements is that it has the element of defence reaction towards uncomfortable feelings from certain memories from the past.

The social unconscious therefore consists of fantasies belonging to wider contexts that the individual is part of (Hopper and Weinberg 2011). For example, Fraher (2004) examines why American commercial airline pilots wanted to carry guns post-9/11. She describes a complex interaction of emotional and unconscious factors: pilots' regression to a heroic, individualistic character; fears of 'foreigners'; and the role of guns in American culture to take action and to restore order. The practice of these fantasies at work 'helped pilots mitigate anxieties arising from a sense of shame for not stopping the hijackings, guilt about the crashes and personal fear of death at the hands of terrorists' (Fraher 2004, p. 574). These are not personal emotions even if they are felt and expressed in that way. The individual desire to carry a gun at work emerges from 'the complex, collusive, emotional, relational, political and historical dimensions affecting the organizational life of commercial pilots' (Fraher 2004, p. 577).

The unmanageable nature of the social unconscious merits attention because it offers the possibility to uncover fantasies that affect the ways systemic power relations are reproduced and become embodied. We suggest that the social unconscious provides a way to see links between power relations and the various emotions evoked around the shared and sometimes unconsciously accepted assumptions of how things should be. Extant literature suggests that fantasies are sustained by facilitating and maintaining knowledge claims that are legitimized by organizational power relations (Willmott 2013). In other words, fantasies are reinforced through the systematization of knowledge claims and by repeatedly protecting and supporting them. For example, fantasies might be sustained

---

[1]　In the post-Freudian literature there is a tendency to spell conscious fantasies and day-dreams as 'fantasy' in order to differentiate them from unconscious psychological activities—'phantasies' (Frosh 2002), but in this study, since the word fantasy is used only in the meaning of unconscious psychological processes, it is spelled as 'fantasy'.

and reinforced by systemic and episodic shaming procedures (Creed et al. 2014). Creed et al. (2014) conceptualized systemic shame as a disciplinary form of power and traced interested members (shamers), who were involved in shaming attempts (making others feel ashamed), to illustrate how shared systemic knowledge around shame was kept intact. Suppressing transgressive behaviours that are potentially dangerous to existing systemic power relations.

We think that the interplay between organizational power relations and the social unconscious is connected to possibilities and limitations regarding organizational learning. First, there are organizational dynamics that underpin how and why individuals share and enact dominant fantasies. These dynamics are guided and supported by emotions (Vince and Gabriel 2011). The interactive dynamics between people, their emotional impulses, and the ways they exercise organizational power relations, are internally influenced. They are fuelled by unconscious fantasy, and subsequently enacted by individuals through everyday organizational relations. Second, interactions between peoples' emotions, the social unconscious, and established power relations maintain characteristic ways of working and become integral to peoples' experiences within it (Frosh 2002), including processes of learning. We further argue that the ways in which power relations hinder or inspire organizational learning are discernible through emotions within and between individuals. These processes consequently determine the ways learning takes place, for example, how and whether it is facilitated, blocked, or resisted (Vince 2001). Our study provides an illustration of these dynamics (see Figure 1).

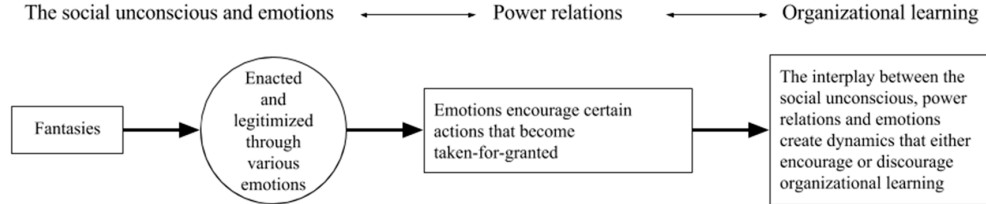

**Figure 1.** The visual depiction of the dynamics between the social unconscious, power relations, emotions, and organizational learning.

## 3. Research Design and Methods

The research approach in this paper is interpretive as we assume that knowledge can be created and understood from the point of view of the research participants who are embedded in the context of the study. Our approach to the data collection and analysis is informed by psychodynamic theory. For instance, we used a visual data collection method of 'photo-elicitation' (Harper 2002; Parker 2009) to improve the potential to generate data reflecting the emotional and the unconscious. We used a pair of photographs, with one depicting an old Soviet workplace and the other a contemporary one. These were introduced to respondents at the end of the interview as a way of contrasting the two sets of stimuli and encouraging respondents to free associate (Sievers 2013) with images that represented past and present experiences of organization. By so doing, we sought to uncover emotions related to both regimes.

This study is taken from a larger empirical data set collected at five organizations in Kazakhstan between February and April 2014. The data were generated from 52 semi-structured, face-to-face interviews (Johnson 2002; Kvale 1996) with an episode of photo-elicitation at the end of each interview (Harper 2002; Parker 2009). The average length of each interview was forty-five minutes and the total amount of data collected from the interviews with photo-elicitation was approximately 1050 transcribed pages. The interviews were conducted both in Russian and Kazakh languages, depending on the research participants' preferences. Twelve interviews were directly translated and transcribed by the first author. For the remaining forty interviews, these were professionally transcribed by transcribers and coded in the original languages. Any parts of these that are cited in this paper were translated into English at a later stage.

The data analysis was designed to bring the respondents' subjective interpretations forward as a source of organizational reality (Corley and Gioia 2004; Seale 1999) and the process of analysis moved

between the emerging categories in the empirical data and the theoretical literature that framed this research (Langley 1999, 2009; Pratt et al. 2006). It consisted of three phases that produced initial codes and categories, second-order themes, and aggregate dimensions (Gioia et al. 2013; Saunders et al. 2009). Initially, similar ideas occurring in the interviews were categorized into different concepts (Gioia et al. 2013; Van Maanen 1979). At this stage, the emergent categories were named using the respondents' own, or similar, descriptive phrases. NVivo 10 (QSR International, Doncaster, Victoria, Australia), a qualitative data analysis software package, was used to undertake the first stage of data coding. This produced around 650 codes, which were then grouped into categories and subcategories (Corley and Gioia 2004).

We discussed, drew and identified a range of emerging themes from our initial analysis. We checked the consistency of these themes with continuous reference back to the interviews, photo-elicitation and diary entries. The process of analysis was not linear but, rather, was carried out recursively, and it is difficult to separate them. We clustered the themes into fewer, overarching aggregate dimensions using our theoretical framing. The aggregate dimensions were explicitly developed to help with identifying the aspects of the social unconscious inherent in the respondents' settings and the emotions surrounding them.

In this paper, we have chosen to focus on one of the aggregate dimensions and to explain it in detail. This was what we saw as a widely shared unconscious mechanism that we term *no room for mistakes*. We see this as a significant aspect of the social unconscious affecting organizational learning in Kazakhstan. We discuss the key aspects of this dimension of organizational learning in detail below and illustrate them with four vignettes. The vignettes help to depict the practicalities of everyday experience in organizations and show key aspects of the social interaction among people in the organizations, as well as their ways of behaving and acting in relation to mistakes at work.

## 4. Analysis

The following four research 'vignettes' illustrate how our aggregate dimension called *no room for mistakes* was expressed and enacted within the organizations we studied. There are three inter-connected elements to this. First, anxiety about making mistakes is a common aspect of people's organizational lives, prompting people to try and avoid, ignore, or disown them. Such anxiety often produces the very thing that it is seeking to avoid (Salecl 2004), which in this context means both making yet more mistakes and (consequently) producing more fear of them. Second, these feelings are further reinforced by the embarrassment and guilt felt when mistakes do come to light because they can take on intense proportions. Third, the intensity of such feelings means that they are either projected onto others, who can represent the mistake and, therefore, take the blame; or they are internalised and reinforce personal feelings of shame. The circularity of this process means that mistakes become embedded in the social order in ways that go beyond ordinary expectations, eventually to become collective fears that inhibit opportunities for learning from mistakes.

### 4.1. Vignette 1

The following vignette was offered by Emma (names have been changed) who was responsible for receiving the orders to produce custom built windows, processing and sending the order to the production department, and shipping them back to the firm's clients. In her narrative, she related how she received an order request, processed it and sent the dimensions of the windows to the production shop. The windows were produced and sent to the client. Upon arrival, the client called back and informed her that the dimensions of the windows were wrong and that they would be sending the windows back. Then Emma opened the purchase order to realize that indeed it was she who had confused the dimensions of the windows and they had been made in the wrong size. In her own words:

> 'I am sitting at my desk in tears not knowing what to do. All the 'wrong' windows would
> be on me now. I would need to pay from my salary. There were so many windows (pause),
> so many, for 10–15 thousand dollars. But [the name of her boss] said: 'Don't worry, we will

find the way to sell them. We will take care of it, nobody will make you pay for this. We will think about it. Maybe there will be a client who will want that size, just write down the sizes'. And she was right. We sold it later to a client who wanted windows in those particular dimensions' (R22).

Emma's experience when encountering her mistake was a feeling of shock and panic. She felt personally responsible for making things right and therefore she felt guilty. She was also shocked that she would have to pay for the rejected windows from her own salary. She cried bitterly at work because she was afraid of the potential financial punishment she was going to receive. She took the mistake into herself. Her boss reassured her by saying that she would not be financially responsible for the windows, that there can be collective and shared feelings around mistakes.

### 4.2. Vignette 2

When his firm failed to win a big contract, the supply chain manager Konstantin, who was leading the bid for the contract, felt direct responsibility for this failure in front of his colleagues. 'Well it did not happen for many reasons. But I should have gotten the contract. I sat at my desk, everybody in the office was in silence. They knew, they understood how bad I felt. This (failure to win the contract) somehow pressured everybody. Everybody sat at their own desks and thought, 'shit!', as if (pause) every one of them felt guilty, as if everyone somehow felt this, did not feel right somehow. It felt as if they felt for me, and what I read in their eyes was: that is fine buddy, there will be more contracts to come. They were with me.' (R38, Supply Chain Manager).

Konstantin feels guilty about the lost contract and projects those same feelings onto others in the office. He is embarrassed and finds it difficult to accept his failure. However, unlike Emma he is unable to contain the embarrassment and disowns such an uncomfortable feeling by projecting it to others. His projection onto others makes him feel easier because he is now not alone in feeling embarrassed for the lost contract. His feelings influence the collective's support of his team, initially in fantasy and then in practice. His internalised fantasy becomes common to everyone in the office.

### 4.3. Vignette 3

'The wrong photograph of the judge from the Higher Court was published. Mistakes are made sometimes. Mistakes are thieves. There was a mistake, and then that issue of the newspaper was published. When I got the phone call the next day, I did not know what to do. I was so...(silent pause). I ended up in the hospital with high blood pressure. There are at least seven or eight people who look at the newspaper before it goes, but nobody noticed it. Then it was in front of them, those people who called from the ministry. None of those seven or eight people are accountable. I am the one who is responsible, because that how it works, they ask the Director about it. Of course, I need to find who made the mistake and punish that person, but still I am the first one to receive the big blow.' (R23, Frank, the Managing Director)

Frank, as the Managing Director of the Company, was accountable to one of the senior government ministries. When he received an accusatory telephone call from the ministry pointing out the mistake in his newspaper, he felt ashamed because he had 'failed to live up to standards of worth in the eyes of others' (Mascolo and Fischer 1996, p. 68). Second, this conceptualization is confirmed by the people from the top who considered this such a serious issue that they felt they needed to point it out. This direct approach is an accepted way to deal with mistakes. Third, after the mistake (i.e., an 'illegal' action) had been pointed out, Frank 'did not know what to do'. It is such an unacceptable and painful experience that it left him perplexed as beforehand he could not have imagined it ever happening. The reality he found himself facing, that, in fact, a very public mistake had been made, appears not to be compatible or comprehensible within the bounds of his work.

Frank went to hospital with high blood pressure. The shame and embarrassment after the phone call grew into anger, and were mixed with his feeling that he was not the only one responsible for the mistake. He feels that he is a victim or a middleman in between the two sides; on the one side, the ministry blaming him and on the other, those who made the error. In addition, as Frank mentioned, 'mistakes are thieves' and he is angry because he fears this mistake will steal his reputation and undermine his credibility in front of the ministry. Frank also reports his intention to find and punish the person who made the mistake.

*4.4. Vignette 4*

> After Mrs. Elton arrives in the office, she 'check(s) all the offices, which door is open which door is closed. Yes, who is in who is not in. Yes, that is how it is, yes. Why isn't he in? What happened? Sick? OK, got sick, what about the other one? Yes, this guy is not in, asleep? Late? If he is late, he knows that he needs to have a word with me.' (R25, Chief Accountant)

Mrs. Elton wants to make sure that everyone is in work and doing what they need to be doing. She is afraid that, if people are not checked, then something might go wrong. She wants to prevent this by punishing those who are late. Her way of thinking has been reinforced by the Deputy Managing Director of this organization who mentioned that 'the collective takes care of itself' (R24) meaning that recourse to their higher managers is not sought in some instances and people clearly know how to handle such situations. The vignette illustrates an example of an unconscious defensive reaction, against the shared belief that everything should be done 'right'. The anxiety of ensuring we are 'doing everything right' makes Mrs. Elton replicate this in her own behaviour, that is, by not allowing mistakes through checking up on all her colleagues. Mrs. Elton is generally suspicious of her colleagues and does not trust them to fulfil their responsibilities. She acts as a controlling mechanism on behalf of the management to make sure that untrustworthy colleagues, who she assumes are incapable of even turning up on time, do their work.

The four vignettes provide narrative examples of how the fear of mistakes becomes embedded in the social order; how these fears are normalised into implicit ways of thinking and working; and how they help to construct the social unconscious of members of a social system in ways that shape people's responses to mistakes. They depict four intersecting 'value-imbued fantasies' (Willmott 2013, p. 55) that have become integral to the social order. First, there is a fantasy that a person (in this case Emma) is going to be punished, that she is going to have to pay for the mistake. Emma's fear is that she will asked to do this literally. Second, there is a fantasy that mistakes are collectively shared when they are too difficult for a leader (in this case Konstantin) to hold onto for himself. Konstantin's anxiety is reduced by the notion that '*they* were with me'. Third, there is the fantasy that others will pay when an individual (in this case Frank) takes responsibility despite his feelings of anger at doing so. Frank takes responsibility because it is his responsibility to do so, but he believes that it is also not his responsibility, that another is responsible, who will be made to pay. Fourth, there is a fantasy that people need to be punished for mistakes before they make them, to reduce the likelihood that they will happen. An individual (in this case Mrs. Elton) takes responsibility for punishing people because they will be culpable in any future mistakes.

## 5. Discussion

Our approach to organizational learning is focused on people's' participation in everyday organizational life and the social processes that inform their shared lived experience. These processes are not necessarily deliberate or conscious, and they are as likely to impede learning as to encourage it. From this perspective, organizational learning is interwoven with organizational power relations as well as unconscious dynamics that can support and undermine people's ability to learn (Vince 2001; Vince and Gabriel 2011). The concept of the social unconscious helps us to capture unconscious dynamics that inform and perpetuate social order by highlighting shared 'anxieties, fantasies, defences, myths, and memories' (Weinberg 2007, p. 312). The social unconscious refers to shared, unconscious

fantasies that are created, sustained, and perpetuated by people in organizations. Fantasies are sustained by facilitating and maintaining knowledge claims that are legitimized by organizational power relations (Willmott 2013) and through prevailing systems. We have identified four interrelated fantasies associated with the fear of mistakes to point out a problem for organizational learning in Kazakhstan.

Fear of mistakes is a common shared emotion in organizations. Such fear prompts people to act with caution, to limit what they aim to do, and if things go wrong, to blame others to rid themselves of such uncomfortable emotions (Vince and Saleem 2004). As our vignettes illustrate, in the Kazakhstani organizations we studied, mistakes are perceived as something unwanted and unwelcome. People feel anxious about making mistakes and their behaviour around actual mistakes generates further anxiety. Their anxiety is reinforced by feelings of embarrassment and guilt, which can become intense. Intense feelings are both internalised and projected onto others, as well as becoming linked to punishment (both of the self and others).

We think that such behaviour has its roots in the politics exercised in the Soviet workplace, and that remains part of people's experience and emotions within the new market system. The feelings and actions accompanying the construct of 'no room for mistakes' and the whole idea of being afraid to make a mistake might well have been inherited as a legacy of '*oblichenie*' (meaning exposure and denunciation). In Soviet workplaces, this was a common way to deal with mistakes. It consisted of two stages, first finding and taking the offender to the Comrades' Court to admonish him/her in front of other colleagues; and second, the 'offender' subsequently was required to undergo corrective measures under the watchful gaze of the collective (Clarke 2007; Kharkhordin 1999).

'Oblichenie' created the Soviet individual as a self-revealing subject in front of the relevant community, which most of the time meant in front of one's own colleagues (Kharkhordin 1999). Throughout the research findings, we have identified two interrelated processes arising from oblichenie, where dynamics between power and emotions turn organizational learning into a challenge. These are denunciating the wrong-doers in public, or shaming them in private. For example, Frank (V3) was the person who received the call about the photo of the wrong judge being published, and was personally blamed for the mistake. Frank's justification for handing down punishment to others comes as the result of his embarrassment and anger, which was rationalized as a need for the prevention of future mistakes. Frank called an emergency meeting and traced the steps back to find out who chose the wrong photo, and then publicly admonished that person in the meeting. Such dynamics make individuals in organizations anxious of any possible mistakes, and as one colleague of Frank's commented: 'I feel the responsibility. If I feel that I did not read the parts of the article, I go through them again. It is important to feel that there will be no scolding and I will not hear unnecessarily from others and nobody will point at me with their finger. It is best not to make a mistake since it comes back and haunts me personally' (R30, Proofreader).

The fear of mistakes in combination with a dressing down in public, in front of colleagues within a meeting, makes employees feel uneasy about meetings. For example, there are times 'when you break out in a cold sweat at the meetings. Is there any mistake I made? Or, what if somebody says something about my work...' (R30, Proofreader). Employees become generally afraid and expect the worst to happen during meetings: 'There are such situations, sometimes there are such meetings after the issue (of the newspaper) is out, and there are people who feel like 'what are the mistakes I made in my writing? I hope I will not be decapitated by the bosses tomorrow' (R28, Journalist).

Blaming is fuelled by the anxiety related to there being no room for mistakes, and the internalized sense of guilt when mistakes occur. Tensions between anxiety and guilt lead to a lack of trust. For example, both Frank and Mrs. Elton believe that some other members of staff are being childish by not being able to manage their time. They micromanage others to avoid mistakes and thereby reinforce the emotions most associated with making them (guilt, embarrassment, and fear). These dynamics align with a study where participants reported that 'in Kazakhstan, you are guilty a priori' (Muratbekova-Touron 2002, p. 223), referring to the ideology of suspicion and control established in the ways people act towards each other.

　　　The organizational dynamics we highlight suggest limitations on organizational capabilities for learning. Instead of learning from mistakes and using them as the means to improve organizational processes, mistakes lead to 'finger pointing' (Clarke 2007). Frank was personally blamed for the mistake by the ministry. Consequently, he vowed to track down the individual who selected the wrong photo for the newspaper and to punish that person. Emma felt her mistake personally as her own 'screw up' and remained anxious until the company sold on the wrongly-measured windows. Konstantin, was not the only person involved during the preparation for working on the potential contract, and upon the realization of the failure he both disowned it by projecting his guilt to others and continued to blame himself. Such instances mean that mistakes are not reviewed to look for constructive ways of overcoming them in the future. Instead, considerable energy is invested in finding and pointing fingers at those individuals who made a mistake.

　　　Our study suggests that internalized habits around mistakes have grown into an organizational practice where people police each other in the service of avoiding them. Some people find themselves playing the role of the 'shamer' (Frank and Mrs. Elton) who victimize others (Creed et al. 2014). Others accept the role of the ashamed. For example, one of the respondents, a journalist, acknowledged that she felt afraid and uncomfortable for being late on her second day after she returned from parental leave: "What shame! Shame in front of the management! What shall I say now? What reason shall I say, and all the way to work I thought about that" (R28, Journalist). Since guilt is an embarrassing and uneasy feeling, to avoid the pain of this, organizational members would sometimes rather lie to cover their backs. The late-for-work journalist claimed that: 'there are times when I am late... sometimes I tell the truth, sometimes, there are times when I lie, there are times when I say the truth as well, that is how it is' (R28, Journalist). She felt embarrassed and was ready to lie as an excuse in front of her colleagues and to avoid being shamed. Such relationships between the shamer and the ashamed are embedded in the organization. Over time, this internalised knowledge seems to settle and does not require the shamers' physical presence. People learn how to feel ashamed without the shamers being there. The shamers (Creed et al. 2014) who police and punish, and the ashamed who are embarrassed and feel guilty, are all caught up in a chain of actions that resist opportunities for learning. These interrelated dynamics create what has been referred to as 'learning inaction' (Vince 2008), which acknowledges an apparent political necessity to resist attempts to learn. The point has been reached where individuals in organizations emotionally or politically become reluctant or choose not to act upon certain things and thus willingly and unwittingly reinforce limitations to organizational learning. In Figure 2 we summarize our proposed conceptualization of the dynamics between the social unconscious, power relations, and organizational learning.

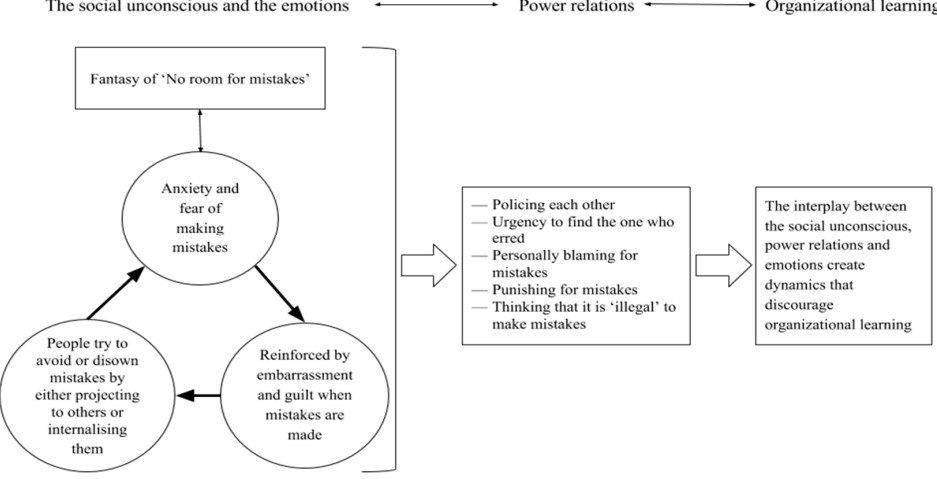

**Figure 2.** The dynamics around 'No room for mistakes' and its impact on organizational learning.

## 6. Conclusions

In this paper, we have addressed a specific aspect of power, emotion and organizational learning in Kazakhstan. This social and political context has provided us with an environment for studying the interplay between emotion and power during processes of learning in organizations. As we identified in the introduction to this paper, our contribution to knowledge arises from employing the idea of the *social unconscious* to show how shared fantasies can sustain and perpetuate power relations and social emotions that constrain learning. Fantasies in relation to mistakes are sustained through blaming and punishing the people who make mistakes; and through feelings of internalised embarrassment and guilt that are enacted through interpersonal relations of shaming and being ashamed.

We have identified three elements to there being 'no room for mistakes' in Kazakhstani organizations. Anxiety about making mistakes has become an integral aspect of people's organizational lives, prompting people to try and avoid, ignore, or disown mistakes. Such anxiety often produces the very thing that it is seeking to avoid, which leads to more mistakes and more fear of mistakes. These feelings are reinforced by the embarrassment and guilt felt when mistakes do come to light, and they can take on intense proportions. The intensity of such feelings means that they are likely to be either projected onto others, who can represent the mistake and, therefore, take the blame; or they are internalised and reinforce personal feelings of shame. Further, embedded emotional dynamics around 'no room for mistakes' inform the organizational power relations. As we have suggested, in the organizations under investigation, policing each other and urgency to find and punish the one who erred seem to be the taken-for-granted and legitimate ways of dealing with mistakes. The circularity of this process means that mistakes become embedded in the social order in ways that go beyond ordinary expectations, eventually to become collective fears that inhibit opportunities for learning from mistakes.

Our identification of a dynamic that we are calling 'no room for mistakes' also implies an interest in what can be done to change such embedded behaviour to more effectively support opportunities for organizational learning. We highlight a practical implication from our study that will need to be an integral aspect of attempts to support organizational learning in post-Soviet Kazakhstan. We think that change will involve practical effort from people within organizations to *make room for mistakes*. The lingering emotional and relational effects of 'oblichenie' have created a contradiction. People have learned how to police each other in the service of avoiding mistakes and (ironically) this behaviour emphasises the anxiety involved in making them. Therefore, they already carry with them a powerful underlying feeling of having made a mistake before it happens. This contradiction connects with existing research on organizational learning that has examined the dynamics of blame (Vince and Saleem 2004) and the nature of the shared anxieties that supports such emotions, as well as how they undermine learning in organizations. One way to support change in people's' perceptions of this contradiction is to accept it as paradoxical, and approach it from a paradox 'mindset'. A paradox mindset depends on 'the extent to which one is accepting of and energised by tensions' (Miron-Spektor et al. 2017), which might include the extent to which such contradictions can be publicly acknowledged rather than publicly avoided. We think that emphasising the contradictory dynamics surrounding mistakes can help to shift the political effects of fantasies that sustain a fear of mistakes.

We are necessarily limited in our ability to make claims beyond the specific organizations we studied, or beyond the specific underlying dynamic that we have identified. However, we do think that our research has begun to pinpoint emotional and political barriers that undermine organizational learning in Kazakhstan. Future research can focus on identifying other collective emotional dynamics (conscious and unconscious) that similarly undermine learning. We think that it will continue to be important engage with the following research questions: what are the underlying emotional and political dynamics that characterize organizations in Kazakhstan and how do they inhibit organizational learning? In addition, it would be interesting to compare the fast growing (e.g., information technology, computer systems) versus stable industries (e.g., transportation, health

care), or multinational versus national organizations might shed light on different learning possibilities. Finally, we argued that social defence reactions taking place in Kazakhstan are the result of abrupt, systemic change. The insights from our study might inform future studies on mistakes, which are not necessarily bound to contexts in change.

**Author Contributions:** Indira Kjellstrand contributed 70% of the paper, and Russ Vince contributed 30% of the paper.

**Conflicts of Interest:** The authors declare no conflict of interest.

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
