# Peer review of "No Room for Mistakes: The Impact of the Social Unconscious on Organizational Learning in Kazakhstan"

_admsci, doi:10.3390/admsci7030027_

Round 1

Reviewer 1 Report

This is an interesting paper that has the potential to make significant contributions to organization studies. In particular, I like the idea of bringing power and emotion as a meaningful attempt to show how organizational learning is inhibited (or enabled). Additionally embedding the learning dynamics within an empirical context of Kazakhstan that experiences considerable social changes, if successful, can substantially advance our understanding of learning. However, I have a strong reservation regarding whether this paper, at its current form, clearly fulfills its promise. Below I present some concerns about conceptual clarity and theoretical contributions.

1. You’ve attempted to situate this paper in an interesting setting—Kazakhstan characterized by institutional tensions between old and new regimes in its transition from a planned economy to a capitalist free market economy. However, I’m not quite sure if your theoretical claims are meaningfully connected to the empirical setting. I may feel this way for several reasons. More specifically:

a)       As this paper relies heavily on social unconscious, fantasies, emotions, and other concepts that have not often been associated with a theory of organizational learning, establishing the conceptual validity and relevancy of these concepts appears to be the first logical step. However, I don’t see strong theoretical justification for using the concepts with respect to organizational learning. For instance, I’m not fully convinced by the claim on p. 9—the concept of the social unconscious helps us to capture unconscious dynamics that inform and perpetuate social order by highlighting shared ‘anxieties, fantasies, defences, myths, and memories.’ What evidence does this paper provide to support this claim? More specifically, how does the concept of social unconscious uncover the causal process in which social order arises in the presence of negative feelings such as anxieties? What is the supporting evidence that shows the relationship between fantasies and learning? How do all the dynamics unfold uniquely in the tension between old and new regimes of Kazakhstan? It would be helpful to offer a more thorough literature review to fully explain how previous lines of work on organizational learning can be informed by the concepts of social unconscious and fantasies and what insights the concepts offer to scholars in the domain of learning and emotions.

b)       I’d love to see more details on your context, Kazakhstan. In Introduction and Discussion, you emphasize the unique nature of Kazakhstan and suggest that there are “unconscious processes that inform collective behavior and organizing in modern Kazakhstan” (p. 1) and that “no room for mistakes” may have its roots in the politics exercised in the Soviet workplace with reference to a legacy of ‘oblichenie’ (p. 9). Thus I feel that readers can better understand your theoretical arguments when the theory is more contextualized. For instance, do you suggest that the concept of social unconscious is best applicable to your context of Kazakhstan, not other countries? If so, why? If not, how does your theory broadly apply to other contexts? What unique aspects of institutional change in Kazakhstan make your theory most suitable to the specific context?

2. I find it difficult to see how your findings drawn from Vignette-based analyses provide convincing evidence to support your theoretical arguments. How do the vignettes demonstrate the interplay between organizational power relations and the social unconscious that is further connected to possibilities and limitations regarding organizational learning? What’s the decisive evidence that reveals the role of emotions in shaping organizational learning? Although I learned from your analysis that mistakes might be connected to negative emotions such as anxiety, it was not entirely clear how such negative feelings and emotions lead to the series of behavioral and organizational consequences you argued in the paper, such as the rise of social order and norms, construction of social unconscious, operation of power relations, emergence of organizational practices, and learning outcomes at the collective level. Providing more specific explanation for each finding would be a welcome improvement.

3. It’d be nice to discuss generality of your findings (see Point 1.b as well). How do your findings apply to other contexts and what are the boundary conditions?

4. It seems there is a disconnect between individual-level emotions associated with mistakes and organization-level learning dynamics. Throughout the paper, you note collective processes in which individual emotions do not stay within an individual; rather mistakes and fantasies go beyond the affected individuals and become “collective fears that inhibit opportunities for learning from mistakes” (p. 11). Individual emotions can also turn into collective emotional dynamics and even shape organizational practices that now guide and direct other organizational members who do not directly experience emotions (p. 12). However, I couldn’t find detailed discussion on collective emotional dynamics (rarely mentioned in the analysis and results). It’d be nice to wrap up in the discussion and conclusion sections by more clearly describing how this paper finally resolves the core issues associated with the collective emotional dynamics and how future research can build upon your findings.

I do hope you find my comments and suggestions useful in improving your paper.

Author Response

(x) I would not like to sign my review report

( ) I would like to sign my review report

English language and style

( ) Extensive editing of English language and style required

( ) Moderate English changes required

(x) English language and style are fine/minor spell check required

( ) I don't feel qualified to judge about the English language and style

Yes

Can be improved

Must be improved

Not applicable

Does the introduction provide   sufficient background and include all relevant references?

( )

(x)

( )

( )

Is the research design appropriate?

( )

( )

(x)

( )

Are the methods adequately   described?

( )

(x)

( )

( )

Are the results clearly presented?

( )

(x)

( )

( )

Are the conclusions supported by   the results?

( )

( )

(x)

( )

Comments and Suggestions for Authors

This is an interesting paper that has the potential to make significant contributions to organization studies. In particular, I like the idea of bringing power and emotion as a meaningful attempt to show how organizational learning is inhibited (or enabled). Additionally embedding the learning dynamics within an empirical context of Kazakhstan that experiences considerable social changes, if successful, can substantially advance our understanding of learning. However, I have a strong reservation regarding whether this paper, at its current form, clearly fulfills its promise. Below I present some concerns about conceptual clarity and theoretical contributions.

(The reviewer’s text is in black and our reply is in blue)

1. You’ve attempted to situate this paper in an interesting setting—Kazakhstan characterized by institutional tensions between old and new regimes in its transition from a planned economy to a capitalist free market economy. However, I’m not quite sure if your theoretical claims are meaningfully connected to the empirical setting. I may feel this way for several reasons. More specifically:

a)       As this paper relies heavily on social unconscious, fantasies, emotions, and other concepts that have not often been associated with a theory of organizational learning, establishing the conceptual validity and relevancy of these concepts appears to be the first logical step. However, I don’t see strong theoretical justification for using the concepts with respect to organizational learning. For instance, I’m not fully convinced by the claim on p. 9—the concept of the social unconscious helps us to capture unconscious dynamics that inform and perpetuate social order by highlighting shared ‘anxieties, fantasies, defences, myths, and memories.’ What evidence does this paper provide to support this claim? More specifically, how does the concept of social unconscious uncover the causal process in which social order arises in the presence of negative feelings such as anxieties? What is the supporting evidence that shows the relationship between fantasies and learning?

In this paper, we investigated the (unconscious) fantasy of ‘no room for mistakes’ which is one example of the social unconscious that informs social order. We use four vignettes to support our theoretical claim. Each vignettes provides an example of the fantasy of ‘no room for mistakes’. For instance, in vignette #3, Frank discusses his feelings and actions around the wrong photo of the judge printed in one of the newspaper issues. He gets, as he feels, an accusatory telephone call pointing out this mistake. Franks reports his feelings as following: ‘When I got the phone call the next day, I did not know what to do. I was so... (silent pause). I ended up in the hospital with high blood pressure.’ (Vignette #3; p9) We think that he felt embarrassed and guilty in front of the ministry people as he ends up at the hospital. Then he argues that ‘Of course, I need to find who made the mistake and punish that person (ibid). We think that this is an example of the emotional dynamics around ‘no room for mistakes’.

As we further argued in the discussions section, the dynamics around this fantasy impact on organizational learning. In the case of Frank, the anxiety around this mistake is reinforced by the feeling of embarrassment, which is further followed by projection. He is projecting it to others because he reports it as someone else’s mistake, and he is just a middleman who ‘receives the big bowl’ (ibid). This is the emotional dynamics around ‘no room for mistakes’. Next, such emotional dynamics enters the political order as an urgency to find the one who erred every time mistakes happen. Moreover, the interplay between emotions and power relations around ‘no room for mistakes’ create certain dynamics such as fear of meetings. As one of Frank’s colleagues claimed: ‘There are such situations, sometimes there are such meetings after the issue (of the newspaper) is out, and there are people who feel like ‘what are the mistakes I made in my writing? I hope I will not be decapitated by the bosses tomorrow’ (page 12; R28, Journalist).  We argue that such dynamics limit the possibility to learn because instead of noting this mistake and making an attempt to avoid it in the future, people in this organization seem to be keen to find the one who erred and punish him/her. We suggest that this is an illustration of the fantasy in action. In the revised version we provided a graphical illustration of these dynamics (Figure 2). We think that this will sufficiently clarify the relationship between the concepts we used.  

How do all the dynamics unfold uniquely in the tension between old and new regimes of Kazakhstan? It would be helpful to offer a more thorough literature review to fully explain how previous lines of work on organizational learning can be informed by the concepts of social unconscious and fantasies and what insights the concepts offer to scholars in the domain of learning and emotions.

This was a helpful remark, and in the revised version we considered this advice and added a paragraph in the introduction part on the previous studies on Kazakhstan and on emotions and organizational learning.

b)     I’d love to see more details on your context, Kazakhstan. In Introduction and Discussion, you emphasize the unique nature of Kazakhstan and suggest that there are “unconscious processes that inform collective behavior and organizing in modern Kazakhstan” (p. 1) and that “no room for mistakes” may have its roots in the politics exercised in the Soviet workplace with reference to a legacy of ‘oblichenie’ (p. 9). Thus I feel that readers can better understand your theoretical arguments when the theory is more contextualized.

For instance, do you suggest that the concept of social unconscious is best applicable to your context of Kazakhstan, not other countries? If so, why? If not, how does your theory broadly apply to other contexts?

We do suggest that the social unconscious is the best applicable to the chosen context because the systemic change it is undergoing at the moment provides an environment rich in emotions and power relations. We suspect that it would be applicable to other post-Soviet environments but we have not studies the distinctive characteristics of other post-Soviet environments. 

What unique aspects of institutional change in Kazakhstan make your theory most suitable to the specific context?

This study does not refer to the theory of institutional change, however we pointed out that undergoing systemic change from the Soviet style management to free market economy provides a useful context for our study.

2. I find it difficult to see how your findings drawn from Vignette-based analyses provide convincing evidence to support your theoretical arguments.

How do the vignettes demonstrate the interplay between organizational power relations and the social unconscious that is further connected to possibilities and limitations regarding organizational learning?

What’s the decisive evidence that reveals the role of emotions in shaping organizational learning?

Although I learned from your analysis that mistakes might be connected to negative emotions such as anxiety, it was not entirely clear how such negative feelings and emotions lead to the series of behavioral and organizational consequences you argued in the paper, such as the rise of social order and norms, construction of social unconscious, operation of power relations, emergence of organizational practices, and learning outcomes at the collective level. Providing more specific explanation for each finding would be a welcome improvement.

Considering this suggestion you made, we provided the graphical illustration of our theoretical frame and our findings in two separate figures. We think that the graphical illustration helps to explain the ways different literature we use are connected.

3. It’d be nice to discuss generality of your findings (see Point 1.b as well). How do your findings apply to other contexts and what are the boundary conditions?

This is an interpretive study therefore we are not imagining generality so much as high contextual relevance. Our findings are bound to the context and even to the organizations we have studied.

4. It seems there is a disconnect between individual-level emotions associated with mistakes and organization-level learning dynamics. Throughout the paper, you note collective processes in which individual emotions do not stay within an individual; rather mistakes and fantasies go beyond the affected individuals and become “collective fears that inhibit opportunities for learning from mistakes” (p. 11). Individual emotions can also turn into collective emotional dynamics and even shape organizational practices that now guide and direct other organizational members who do not directly experience emotions (p. 12). However, I couldn’t find detailed discussion on collective emotional dynamics (rarely mentioned in the analysis and results). It’d be nice to wrap up in the discussion and conclusion sections by more clearly describing how this paper finally resolves the core issues associated with the collective emotional dynamics and how future research can build upon your findings.

We have considered this suggestion and now the focus of the study stresses the relational aspect of learning i.e. learning processes are now conceptualised not separately as individual or organizational but rather happening within and between individuals:  ‘The tension between old and new regimes provides an environment that is rich both in emotion and power/politics, and it offers an opportunity to study how the interplay between emotion and power affect the learning processes within and between during individuals as well as organizational learning and organizational attempts to learn.’ (p1)

Reviewer 2 Report

This paper explores makes an interesting contribution to the literature by exploring the role of the ‘social unconscious’ in organizational learning.  The qualitative inductive method is appropriate for studying the research question at hand and the author(s) do a good job clearly describing the research design and methods.  The vignettes are interesting and provide greater texture of the findings for the reader.

A few comments on how the paper might be further strengthened are included below:

1 – In the introduction the author(s) indicate that the research context “offers an opportunity to study the interplay between emotion and power during individual and organizational attempts to learn”.  More description of how this is a practical problem that needs to be addressed would be helpful to bring clarity to the problem that is under investigation i.e. what do challenges to learning look like specifically in Kazakhstan and how does this connect to broader challenges identified in the organizational learning literature?

2 – The author(s) state that the practical purpose of the paper is to “improve our knowledge of the social and political context of organizational learning in post Soviet Kazakhstan through understanding unconscious dynamics that both inform and undermine attempts to learn”.  You revisit this purpose in the implications of your research in the discussion. While your methodology is appropriate for richly exploring a grounded context rather than for generalizing, it seems that the purpose of the paper and the conclusions are a bit more modest than your methods require.  It would be good to indicate what insights from your work might inform future research and current practice in this domain in other contexts and then work to identify those in your discussion i.e. are there similarly politically charged contexts where these findings might be relevant to advance our understanding organizational learning – for example, government organizations?  Are there other cultural contexts where these insights might be useful - for example, an organization where ‘expert’ power is strong, such as a hospital?  How does this research inform similar contexts where there might be a culture of ‘blame’ or a ‘lack of trust’ – I suspect that similar blocks to learning might be observed in other contexts where there is ‘no room for mistakes’.  Drawing out those barriers to learning might be useful to inform research in those organizations. What might future research focus on to elaborate and build on the findings from this study that inform your specific research context?

3 – The use of a figure would help to clarify the conceptual framework you are developing at the outset of the paper (prior to the section on research design and methods).  The many relationships described and the richness of the qualitative themes leave the reader to do a lot of work to clarify what you are studying and what you discovered.  I found myself drawing a picture to sort out these relationships and findings.  Visually depicting how you are exploring the relationship between organizational learning, power and the social unconscious and then revisiting that figure in your findings would help to clarify what connections you are exploring and how you understand them differently after conducting this study.  Figure 1 would be what we currently know/don’t know based on the literature and Figure 2 would include insights from your research i.e. mapping on the ‘four fantasies’ that comprise your theme ‘no room for mistakes’ as elements of the social unconscious and visually depicting their relationship with organizational learning and power.

4 – Is it helpful to use the terms individual and organizational learning when you have described learning as both ‘taking place in individuals’ minds’ and as a ‘process of participation and interaction’?   It seems that you are focused on learning that happens within and between individuals in organizations by virtue of the data that you have gathered and your focus on the social unconscious, rather than on an organization level phenomenon.  Clarifying this in the section “Organizational Learning, Power and the Social Unconscious” will help with situating your contribution within the organizational learning literature.

5 – The author(s) are clear on the contribution OF the literature for understanding their findings.  More specifically describing how they have contributed TO the organizational learning literature would strengthen the paper’s conclusions.

I very much enjoyed reading the paper!  I hope that you take these comments in the spirit that they were intended, that is, as constructive remarks to help you strengthen the work.  Best of luck with the paper.

Author Response

English language and style

( ) Extensive editing of English language and style required
( ) Moderate English changes required
(x) English language and style are fine/minor spell check required
( ) I don't feel qualified to judge about the English language and style
Yes      Can be improved        Must be improved       Not applicable
Does the introduction provide sufficient background and include all relevant references?  ( )         (x)        ( )            ( )
Is the research design appropriate?   (x)        ( )         ( )         ( )
Are the methods adequately described?        (x)        ( )         ( )         ( )
Are the results clearly presented?      ( )         (x)        ( )         ( )
Are the conclusions supported by the results?           ( )         (x)        ( )         ( )
Comments and Suggestions for Authors
This paper explores makes an interesting contribution to the literature by exploring the role of the ‘social unconscious’ in organizational learning.  The qualitative inductive method is appropriate for studying the research question at hand and the author(s) do a good job clearly describing the research design and methods.  The vignettes are interesting and provide greater texture of the findings for the reader.

(AUTHOR’S REPLY AT THE BOTTOM IN BLUE TEXT)

A few comments on how the paper might be further strengthened are included below:

1 – In the introduction the author(s) indicate that the research context “offers an opportunity to study the interplay between emotion and power during individual and organizational attempts to learn”.  More description of how this is a practical problem that needs to be addressed would be helpful to bring clarity to the problem that is under investigation i.e. what do challenges to learning look like specifically in Kazakhstan and how does this connect to broader challenges identified in the organizational learning literature?

2 – The author(s) state that the practical purpose of the paper is to “improve our knowledge of the social and political context of organizational learning in post Soviet Kazakhstan through understanding unconscious dynamics that both inform and undermine attempts to learn”.  You revisit this purpose in the implications of your research in the discussion. While your methodology is appropriate for richly exploring a grounded context rather than for generalizing, it seems that the purpose of the paper and the conclusions are a bit more modest than your methods require.  It would be good to indicate what insights from your work might inform future research and current practice in this domain in other contexts and then work to identify those in your discussion i.e. are there similarly politically charged contexts where these findings might be relevant to advance our understanding organizational learning – for example, government organizations?  Are there other cultural contexts where these insights might be useful - for example, an organization where ‘expert’ power is strong, such as a hospital?  How does this research inform similar contexts where there might be a culture of ‘blame’ or a ‘lack of trust’ – I suspect that similar blocks to learning might be observed in other contexts where there is ‘no room for mistakes’.  Drawing out those barriers to learning might be useful to inform research in those organizations. What might future research focus on to elaborate and build on the findings from this study that inform your specific research context?

3 – The use of a figure would help to clarify the conceptual framework you are developing at the outset of the paper (prior to the section on research design and methods).  The many relationships described and the richness of the qualitative themes leave the reader to do a lot of work to clarify what you are studying and what you discovered.  I found myself drawing a picture to sort out these relationships and findings.  Visually depicting how you are exploring the relationship between organizational learning, power and the social unconscious and then revisiting that figure in your findings would help to clarify what connections you are exploring and how you understand them differently after conducting this study.  Figure 1 would be what we currently know/don’t know based on the literature and Figure 2 would include insights from your research i.e. mapping on the ‘four fantasies’ that comprise your theme ‘no room for mistakes’ as elements of the social unconscious and visually depicting their relationship with organizational learning and power.

4 – Is it helpful to use the terms individual and organizational learning when you have described learning as both ‘taking place in individuals’ minds’ and as a ‘process of participation and interaction’?   It seems that you are focused on learning that happens within and between individuals in organizations by virtue of the data that you have gathered and your focus on the social unconscious, rather than on an organization level phenomenon.  Clarifying this in the section “Organizational Learning, Power and the Social Unconscious” will help with situating your contribution within the organizational learning literature.

5 – The author(s) are clear on the contribution OF the literature for understanding their findings.  More specifically describing how they have contributed TO the organizational learning literature would strengthen the paper’s conclusions.

I very much enjoyed reading the paper!  I hope that you take these comments in the spirit that they were intended, that is, as constructive remarks to help you strengthen the work.  Best of luck with the paper.

REPLY

This was a very helpful review and we took into consideration every single suggestion you made and revised our paper accordingly. Thank you so much for your valuable recommendations.  

Reviewer 3 Report

Suggest incorporating more discussion about organizational and meta-cultural (c.f. Schein) culture and influences.  These are very relevant to social unconscious as they are part of the underlying assumption system or cultural DNA as Schein and others suggest.  Also reference to concept of underlying social cognition (see Jackendorf concepts of culture and language) and underlying systems (e.g. Joanne Martin on culture) and structuration theory and culture could be applied.

Author Response

(x) I would not like to sign my review report

( ) I would like to sign my review report

English language and style

( ) Extensive editing of English language and style required

( ) Moderate English changes required

(x) English language and style are fine/minor spell check required

( ) I don't feel qualified to judge about the English language and style

Yes

Can be improved

Must be improved

Not applicable

Does the introduction provide   sufficient background and include all relevant references?

(x)

( )

( )

( )

Is the research design appropriate?

( )

(x)

( )

( )

Are the methods adequately   described?

( )

(x)

( )

( )

Are the results clearly presented?

(x)

( )

( )

( )

Are the conclusions supported by   the results?

(x)

( )

( )

( )

Comments and Suggestions for Authors

Suggest incorporating more discussion about organizational and meta-cultural (c.f. Schein) culture and influences. These are very relevant to social unconscious as they are part of the underlying assumption system or cultural DNA as Schein and others suggest. Also reference to concept of underlying social cognition (see Jackendoff concepts of culture and language) and underlying systems (e.g. Joanne Martin on culture) and structuration theory and culture could be applied.

REPLY:

Thank you very much for your thoughts on our manuscript. We reflected on the points you raised for us but we do not think that incorporating either the literature on organizational and meta-culture or the literature on social cognition and underlying systems would help our current research. We argued our reason by comparing these literatures with the social unconscious. 

Differences between Organizational Culture and the Social Unconscious

1.    Looking from the organizational culture literature, the scholars in the area argues about different cultures each organization develops, and the social unconscious concerns the social level unconscious that can be inherent in several organizations in a given context.

2.    One can further argue that what we study may belong to Kazakhstani culture or Post-Soviet culture which might be true, as ‘culture is the set of important understandings that members of a community share in common’ (Sathe, 1985:6). Furthermore, our findings may illustrate certain espoused values, which are defined as ‘the articulated, publicly announced principles and values that the group claims to be trying to achieve, such as ‘product quality’ or ‘price leadership’ (Schein, 2004:13)’. It may even cover the habits of thinking, mental models and linguistic paradigms, which represent ‘the shared cognitive frames that guide the perception, thoughts, and language used by the members of a group and taught to new members in the early socialization process’ (Schein, 2004:13). Still both shared cognitive models as well as the espoused values are conscious efforts of the mind which are different from the unconscious aspects happening at the workplace.   

3.    Moreover, the social unconscious, as many other types of unconscious reactions, has to do with defense mechanisms. As Frosh puts it, defence ‘denotes a supposed attempt by the other to stave off an attack on her or himself, a process of denial, particularly of uncomfortable emotional truth’ (Frosh, 2002:26). The fantasies we presented in this paper are mostly defensive reactions that have been developed under the previous regime and lingering in today’s organizations in Kazakhstan. There exists an element of protection of the self, against unconscious (and frequently uncomfortable) knowledge, and as we argue such unconscious knowledge is the inherent fear of mistakes that were punishable during the Soviet times. The unconscious discussed in the literature of culture and language is more relates to the unconscious that allows the people automatically, effortlessly and intuitively react to situations and direct their behaviour (Epstein, 1994; Jackendoff, 2007). For instance, shaking hands and knowing when to do that during conversation; knowing how to dress to a funeral (Schein, 2004); changing the gear while driving (Jackendoff, 2007).        

Similarities between culture and the Social Unconscious

1.    Structural stability - culture is frequently defined by certain values, symbols and shared meaning which implies its stably ingrained in the structure (Martin, 2002). It is the same with the SU, it is ingrained in what people do and enacted in taken-for-granted actions (Frosh, 2002).

2.    Depth and breadth - ‘culture is the deepest, often unconscious part of a group and it is therefore less tangible and less visible’ (Schein, 2010:16). The SU is unconscious and non-tangible that is why we used the photo-elicitation to uncover the hidden emotions because the unconscious is discernible through those emotions.

Social Cognition VS the Social Unconscious

Social cognition can contribute to the literature on organizational learning and there is a broad array of literature in cognitive studies of organizational learning (XXX- copy from the literature review of the thesis). Social cognition ‘focuses on the individual’s knowledge, understanding, and ability in social/cultural contexts more than the structure of the culture as a whole’ (Jackendoff, 2007:146). However, it studies the overt processes whereas the social unconscious studies covert processes that are not readily visible. Therefore we used psychodynamics and ‘illustrated’ it through emotions felt and displayed by our respondents.

References

Epstein, S. (1994). Integration of the cognitive and the psychodynamic unconscious. American psychologist, 49(8), 709.

Frosh, S. (2003). Key concepts in psychoanalysis. NYU Press.

Jackendoff, R. (2007). Language, consciousness, culture: Essays on mental structure (Vol. 2007). MIT Press.

Sathe, V. (1985). Culture and related corporate realities: Text, cases, and readings on organizational entry, establishment, and change. Richard D Irwin.

Schein, E. H. (2010). Organizational culture and leadership (4th edition) John Wiley & Sons.

Schein, E. H. (2004). Organizational culture and leadership (3rd edition). California, USA: Jossey-Bass.

Round 2

Reviewer 1 Report

Please refer to the attached comments.
